# Extracellular Vesicles in Diabetic Cardiomyopathy—State of the Art and Future Perspectives

**DOI:** 10.3390/ijms25116117

**Published:** 2024-06-01

**Authors:** Przemysław Zygmunciak, Katarzyna Stróżna, Olga Błażowska, Beata Mrozikiewicz-Rakowska

**Affiliations:** 1Faculty of Medicine, Medical University of Warsaw, 02-091 Warsaw, Poland; zygmunciakprzemyslaw@gmail.com (P.Z.);; 2Department of Endocrinology, Centre of Postgraduate Medical Education, Marymoncka St. 99/103, 01-813 Warsaw, Poland

**Keywords:** diabetic cardiomyopathy, DCM, extracellular vesicles, exosomes, micro-RNA, diabetes, heart failure

## Abstract

Cardiovascular complications are the most deadly and cost-driving effects of diabetes mellitus (DM). One of them, which is steadily attracting attention among scientists, is diabetes-induced heart failure, also known as diabetic cardiomyopathy (DCM). Despite significant progress in the research concerning the disease, a universally accepted definition is still lacking. The pathophysiology of the processes accelerating heart insufficiency in diabetic patients on molecular and cellular levels also remains elusive. However, the recent interest concerning extracellular vesicles (EVs) has brought promise to further clarifying the pathological events that lead to DCM. In this review, we sum up recent investigations on the involvement of EVs in DCM and show their therapeutic and indicatory potential.

## 1. Introduction

Diabetes mellitus (DM) is an escalating problem worldwide. Its prevalence is steadily increasing, with 578 million people projected to suffer from it in 2030, accounting for 25% growth since 2019 [1]. Unhealthy dietary habits, a sedentary lifestyle, and the growing prevalence of obesity are just a few reasons for the young onset of DM [2,3]. Its combination with the current demographic alterations including population aging drives DM’s growth in prevalence [2,3]. The longer the time for which one’s blood sugar level is abnormal, the more pronounced the risk of developing diabetic complications is, among which, cardiovascular disease (CVD) is the most common [4].

Indeed, diabetes is a well-established risk factor for numerous cardiovascular diseases, including diabetic cardiomyopathy (DCM) [5,6,7]. The link between increased blood sugar levels and dysfunction of cardiac muscle without other complicating conditions such as hypertension or coronary artery disease (CAD) was first described over 50 years ago [8]. This manuscript set the foundations for decades of further research on DCM. Even now, the pathomechanisms of the illness remain incompletely elucidated. In fact, the existence of the disease is still being discussed [9]. However, this has not stopped research considering DCM and, despite obstacles such as the lack of a clear definition of the disease, the interest in the matter thrives. Several groups of researchers have tried to propose universally accepted diagnostic criteria; nevertheless, a consensus is yet to be reached. The least strict definition requires left ventricular dysfunction in the absence of CAD in patients with diabetes [10]. Other publications are more rigorous, naming hypertension and cardiovascular conditions, such as valvular abnormalities, as additional exclusion criteria [11,12,13]. Nonetheless, there is an argument to be made that hypertension under appropriate antihypertensive treatment should not constitute an exclusion criterion for diagnosis [14]. In either case, the much-expected universal diagnostic criteria for the condition remain unattainable.

Diabetes is an independent heart failure risk factor, and its appearance increases the likelihood of the development of the condition by 1.5–2-fold [15,16]. In a population-based study from Olmsted County, the community prevalence of DCM was estimated to be 1.1% [17]. DCM, described in the study as left ventricular dysfunction in the absence of CAD, valvular disease, or hypertension, was diagnosed in almost 17% of the diabetic patients, and diastolic dysfunction was the most common phenotype [17]. Further evidence for the high prevalence of left ventricular dysfunction among diabetic patients comes from a meta-analysis solely inspecting T2DM patients, where diastolic dysfunction appeared in 48% of the hospitalized T2DM population and 35% of the adult T2DM population [18]. Importantly, a similar prevalence for both men and women was described [18]. It is noteworthy that these studies may not show the real spread of the condition, since the first study only included patients aged 45 years or older, and the second study excluded type 1 diabetes mellitus (T1DM) patients [17,18]. A more thorough investigation is required to understand the epidemiology of DCM better.

Currently, two prevailing theories describe the disease’s pathophysiology, the single and the double clinical phenotype. The single clinical phenotype defines DCM as a continuum from subclinical myocardial changes through diastolic and systolic cardiac muscle dysfunction to heart failure (HF). The essence of the double clinical phenotype theory is the distinction between diastolic and systolic dysfunctions progressing into two separate conditions: heart failure with preserved ejection fraction (HFpEF) or reduced ejection fraction (HFrEF), respectively [19,20]. A single clinical phenotype describes DCM as a gradual disease of two to six steps [20]. This vision is consistent with findings from echocardiography and cardiac magnetic resonance (CMR) [21,22]. However, some studies suggest that the progression from HFpEF to HFrEF is linked with CAD and/or aging rather than diabetes itself [23]. Additionally, some evidence suggests that the pathogenesis of myocardial structural changes in HFpEF and HFrEF differs [24,25]. In HFpEF, initial dysfunction appears in the endothelium, whereas cardiomyocytes are the starting point for HFrEF [24]. Noteworthily, the presentation of patients with T1 and T2DM is not identical, with systolic dysfunction being more common in the former and diastolic dysfunction being linked with the latter [19,26].

Our understanding of the detrimental effects of hyperglycemia on the cardiac muscle has recently expanded with the investigation of extracellular vesicles (EVs). Overlooking DCM during potentially reversible stadiums constitutes the need to find new non-invasive diagnostic and therapeutic methods to prevent the development of the disease. In this review, we explore the current research on DCM in both type 1 and type 2 DM, the involvement of EVs in its pathogenesis, and, from this, the derivation of their diagnostic and therapeutic potential.

## 2. Materials and Methods

An extensive literature search using a combination of four major medical literature databases, Medline, Scopus, Google Scholar, and Embase, was performed. The search considered all full-text articles that were available up to April 2024. No restrictions were applied to the study type or design. The search included the following keywords: “diabetic cardiomyopathy”, “extracellular vesicles”, “exosomes”, “heart failure”, “micro-RNA”, and “miRNA”.

## 3. Extracellular Vesicles

Recent years have seen growing interest in EVs worldwide in preclinical and clinical research. The term EV is the generic name for particles unable to replicate enveloped in a lipid bilayer [27]. EVs are not solely composed of lipid membrane [28]. Their composition includes various proteins, nucleic acids, and lipids which they deliver to specific cells, influencing various intracellular processes [28]. Classically, three groups of EVs are distinguished: exosomes—EVs produced as a result of exocytosis of the multivesicular body (MVB); microvesicles (MVs)—EVs generated via direct budding of the plasma membrane (PM); and apoptotic extracellular vesicles (ApoEVs)—being the result of cellular disassembly during apoptosis [28,29]. This classification, however common, is not recommended by the International Society for Extracellular Vesicles, since it creates inadequate expectations of distinctive biogenesis for each group [27]. Since the EV family is heterogeneous and a consensus on the specific subtype markers is yet to be reached, the nomenclature of EVs is an ongoing problem [27]. Table 1 shows the current terminology of EVs based on several features.

## 4. Diabetic Cardiomyopathy (DCM)

The bioinformatics analysis of the myocardial gene expression has shown that DCM was associated with substantial alterations in genes involved in immune response, metabolism, and inflammation-related pathways [33]. However, the upregulation of these routes is rather the effect of early changes in the cardiac tissue. The initial phase of the disease is rather short and asymptomatic [20]. However, intracellular hyperglycemia and increased fatty acid oxidation (FAO) impair mitochondrial function, thus leading to reactive oxygen species (ROS) and reactive nitrogen species (RNS) formation and putting cells in a state of oxidative distress [19,20,34,35]. This state is the key element of diabetes-induced cellular damage [36]. Importantly, intracellular hyperglycemia directly triggers microvascular injury, whereas macrovascular and cardiomyocyte damage is rather connected with insulin resistance and elevated FAO [37]. This difference may be explained by the lack of the ability to regulate glucose transport by several cells (namely neurons and endothelial, mesangial, and Schwann cells) [36]. Other cells such as cardiomyocytes react to hyperglycemia by translocating glucose transporter type 4 (GLUT4) from their cellular membranes [38,39]. The decrease in intracellular glucose oxidation is compensated by elevated insulin stimulation and consequently increased fatty acid translocase (FAT) exposition on the myocardial membrane, translating to enhanced fatty acid intake and oxidation [39,40]. Nevertheless, unbalanced glucose or fatty acid uptake results in an intracellular shift in glucose metabolism from glycolysis to alternative pathways including polyol, hexosamine, and PKC signaling routes [34,37,41,42]. Both processes are also involved in non-enzymatic processes such as glycation, which induces advanced glycation end-product (AGE) generation [37,42,43]. The cellular effect of this metabolic switch is one of the mechanisms of the cellular damage seen in DCM.

The polyol pathway, also known as the sorbitol pathway, is particularly detrimental in the cells that do not control glucose uptake via insulin stimulus (e.g., endothelium, neurons) [44,45]. As a result of this process, cells experience a hyperglycemic hypoxic state, which leads to neuropathy and endothelial dysfunction [44,45]. Additionally, the crucial enzymes for this pathway—aldose reductase and sorbitol dehydrogenase—indirectly increase oxidative stress by decreasing the availability of their cofactors, nicotinamide adenine dinucleotide phosphate (NADPH) and nicotinamide adenine dinucleotide (NAD+), which translates to reduced antioxidant glutathione (GSH) action and elevated ROS production [34].

The hexosamine pathway is also augmented in diabetic conditions. The glucose metabolism through this pathway leads to UDP (uridine diphosphate) N-acetyl glucosamine buildup, which may be a donor of N-acetyl glucosamine to transcription factors inducing pathological gene expression [36]. For example, enhanced use of the hexosamine pathway promotes transforming growth factor-β1 (TGFβ1) and plasminogen activator inhibitor-1 (PAI-1) expression, inducing diabetes-related vascular injury [46]. The same molecular process leads to endothelial nitric oxide synthase (eNOS) downregulation, further impairing endothelial function [47]. Moreover, the hyperglycemia-induced hexosamine pathway is the driving force of calcium dyshomeostasis in cardiomyocytes [48]. High-glucose treatment of these cells reduces the expression of sarcoendoplasmic reticulum Ca(2+)-ATPase 2a (SERCA2a), which translates to prolonged calcium transience and thus impaired myocardial relaxation [48]. The hexosamine-pathway-induced O-GlcNAcylation of the B-cell lymphoma 2 (BCL-2)-associated agonist of cell death (BAD) was also found in cardiomyoblasts [49]. High-glucose treatment of cardiomyoblasts induced this pathway, leading to increased BAD-BCL-2 dimerization and, therefore, apoptosis [49]. Comparable outcomes were also seen in high-fat-fed rats [50]. Finally, increased glucose metabolism through this pathway has similar effects on GSH as the former, leading to its depletion and increasing cellular oxidative stress [51].

The non-enzymatic process of glycation of proteins, lipids, and nucleic acids leads to the accumulation of AGEs, which are also important in DCM pathogenesis. AGEs are responsible for the regulation of gene expression and influence the reception of extracellular stimuli [52,53]. Additionally, AGEs might interact with receptors for advanced glycation end products (RAGEs), which translates to endothelial damage, a further increase in oxidative stress, and most importantly, the induction of inflammation in the diabetic heart [54]. Indeed, the activation of RAGEs stimulates the signaling of nuclear factor kappa B (NF-κB)-dependent proinflammatory mediators including tumor necrosis factor alpha (TNF-α), interleukin (IL)-1, IL-6, and RAGE itself, initiating a vicious cycle [55]. What is more, the injection of high-molecular-weight (HMW-)AGEs induces cardiac dysfunction in vivo [56]. The hearts of the rats treated with HMW-AGEs were characterized by elevated mass as well as decreased contractility [56]. The microscope analysis revealed cardiomyocytes’ concentric hypertrophy, whereas ultrastructural examination has shown disrupted cellular architecture, decreased mitochondrial density, and functionality [56]. What is more, other intracellular alterations due to HMW-AGEs’ injection such as disrupted Ca^2+^ cycling and decreased myofilament function were described [56]. Similar results considering disrupted Ca^2+^ influx and reduced contractility were observed even after a short period of HMW-AGE treatment [57]. What is more, comparable outcomes were described even after preconditioning the cardiomyocytes with anti-RAGE antibodies, which indicates that the aforementioned effects are the consequence of AGEs’ actions rather than AGE-RAGE activation [58].

Lastly, the myocardial triglyceride content is elevated in type 2 diabetic patients [59]. Cardiomyocytes are not designed to store lipids; therefore, intermediates of FAO are directed toward diacylglycerol (DAG) and ceramide generation [60]. DAG is a cofactor of PKC activation—the last crucial metabolic pathway augmented in hyperglycemic conditions [36,60]. PKC overstimulation translates to several pathological mechanisms of DCM such as vascular injury (eNOS downregulation and endothelin-1 upregulation), fibrosis (increased deposition of collagen and fibronectin via TGFβ overexpression), and inflammation (NF-κB-dependent cytokine release) [36]. The activation of PKC promotes p66^Shc^ phosphorylation, stimulating apoptosis and increasing mitochondrial H_2_O_2_ production, which further increases cellular oxidative stress [61]. Moreover, transient PKC stimulation is also linked with disrupted calcium handling in cardiomyocytes [62]. This alteration is the consequence of cytoskeletal remodeling and T-tubule damage observed in cells with increased PKC activation [62]. Myocardial PKC stimulation also leads to angiotensin-converting enzyme (ACE) gene upregulation [35] This is believed to be one of the causes of congestive heart failure due to cardiomyocytes’ hypertrophy and cardiac fibrosis triggered by ACE induction [35]. Lastly, PKC action in fibroblasts results in elevated collagen deposition and, thus, fibrosis [25].

As noted above, cardiomyocytes of diabetic patients have increased fatty acid metabolism. However, the activation of PKC is not the only catastrophic effect of this metabolic switch. Patients with metabolic syndrome have an increased sterol-regulatory element-binding protein (SREBP)-1c/peroxisome proliferator-activated receptor-γ (PPARγ) pathway, which is correlated with decreased ejection fraction (EF) and elevated lipid accumulation [63]. High levels of fatty acids and elevated PPAR-α are associated with uncoupling protein 3 (UCP-3) overexpression [64]. Uncoupling is a process reducing mitochondrial ATP production which correlates with the impaired contractility of the heart muscle [65]. Additionally, heart muscle insufficiency is not the only change initiated by increased lipid metabolism. In vitro studies of atrial tissue indicate that diabetic heart mitochondria are characterized by a decreased capacity to oxidize fatty acids with increased H_2_O_2_ generation, which shows another mechanism of oxidative stress induction [66]. Additionally, cardiomyocyte loss may be initiated by the process called lipoapoptosis where palmitate toxicity, ER stress, inflammation, ceramide, and diacylglycerol formation play a major role [65].

Taken together, the oxidative stress initiated by hyperglycemia and increased FAO leads to mitochondrial damage which induces cellular oxidative stress. This state activates additional glucose metabolism pathways which, combined with the effects of lipotoxicity, further amplify ROS generation and specific cellular alterations leading to endothelial, neuronal, and cardiomyocyte damage. All of these processes mark the middle stage of DCM, where the first conventional echocardiological findings and symptoms appear [20]. In the final stage of DCM, the cellular damage and fibrosis further accelerate, which translates to decreased EF and severe exercise intolerance [20].

## 5. Involvement of Extracellular Vesicles in the Pathogenesis of Diabetic Cardiomyopathy

### 5.1. Cardiomyocyte Death and Hypertrophy

Cardiomyocyte apoptosis leads to the weakening of the structure of the heart muscle. Cardiomyocyte remnants compensate for this loss by the induction of hypertrophy [10,67]. In DCM, the balance between apoptosis and cardiomyocyte homeostasis is disturbed (Figure 1). Exosomal miRNAs affect various cellular processes, mediate intercellular communication, and influence cell survival. One of the most significant signaling routes responsible for regulating apoptosis and autophagy is the Hippo pathway. Mammalian sterile 20-like kinase 1 (Mst1) plays a crucial role in this process [68]. In vivo studies have shown worsening cardiac function and increased insulin resistance with the excessive expression of Mst1 [68]. Elevated levels of Mst1 protein were also found in exosomes derived from cardiac endothelial cells. These exosomes may be taken up by cardiomyocytes, resulting in reduced autophagy and enhanced apoptosis under high-glucose conditions [69]. Exosomes with the detrimental Mst1 protein also inhibit GLUT4 membrane translocation in cardiomyocytes during hyperglycemia. This may lead to insulin resistance and consequently contribute to cardiomyopathy conditions [69].

There is evidence that exosomal miRNAs are involved in the inhibition and induction of cardiomyocyte hypertrophy [70,71]. Another mechanism leading to cardiac hypertrophy and ultimately HF is the persistent stimulation of the local Renin–Angiotensin System (RAS) within the heart [72]. As mentioned before, activation of the PKC pathway leads to increased ACE gene expression, which translates to elevated local angiotensin II (ATII) concentrations [35,72]. Some investigations imply that angiotensin II directly induces cardiomyocyte hypertrophy [73,74]. Studies in neonatal rat cardiomyocytes demonstrate a paracrine mechanism between cardiac fibroblasts (CFs) and cardiomyocytes in which Ang II induces pathological cardiac hypertrophy [72]. Angiotensin II increased exosome release by activating AT1R and AT2R receptors in CF cultures [72]. It has been shown that cardiac fibroblasts produce pro-hypertrophic substances, with cardiac fibroblast exosomes being the sole carrier [75,76]. In addition, exosomes derived from cardiac fibroblasts have been found to cause the upregulation of RAS, resulting in cardiomyocyte hypertrophy [72]. The suppression of exosome production inhibits Ang II-induced cardiac enlargement [72]. Among extracellular vesicles associated with miRNAs, miR-21-3p stands out [75]. miR-21 is elevated in response to hyperglycemia [77]. Its pharmacological inhibition reduces the development of cardiomyopathy in an Ang II-induced animal model by inhibiting cardiomyocyte hypertrophy. This process is caused by the downregulation of SORBS2 and PDLIM5 gene expression, which are regulated by miR-21 [75].

### 5.2. Endothelial Damage

Endothelial cells (ECs) are essential for the survival of cardiomyocytes because they carry oxygenated blood and transmit protective signals [78]. However, in DCM, several negative processes affect EC homeostasis (Figure 2). Reduced capillary density in the myocardium, known as microvascular rarefaction, and impaired angiogenesis are considered the main symptoms of cardiovascular disease caused by diabetes, where endothelial dysfunction and death occur [79,80]. Of great importance in the process are miR-126, miR-320, and miR-503 as miRNAs involved in the regulation of endothelial function and angiogenesis in type 2 diabetes [81]. Caporali et al. demonstrated the role of miRNA in endothelial defects caused by diabetes [82]. Cdc25 is a direct target of miR-503 [82]. It is regulated by high glucose levels. This leads to the inhibition of endothelial proliferation and angiogenesis [82]. The transfer of miRNA-503 from EC-derived EVs reduced pericyte migration and proliferation, leading to reduced angiogenesis and vascular permeability via the inhibition of VEGFA (vascular endothelial growth factor A) and EFNB2 (Ephrin B2) [82]. This process occurs through the activation of the p75 neurotrophin receptor (p75NTR), which leads to the activation of the NF-kB (nuclear factor kappa-light-chain-enhancer of activated B cells) pathway. The expression of miR-503 in ECs is increased by the NF-kB pathway binding to the miR-503 promoter. Additionally, NF-kB activates Rho kinase, which induces the creation of microparticles containing miR-503. These microparticles effectively transfer miR-503 from endothelial cells to vascular pericytes. miR-503 disrupts the expression of VEGFA and EFNB2, leading to the inhibition of angiogenesis and vessel integrity [82]. Studies have confirmed that the transport of miR-320 from cardiomyocytes to endothelial cells depends on extracellular vesicles [81]. Furthermore, elevated levels of miR-320 inhibit angiogenesis [81]. This is due to the reduced proliferation, migration, and tube formation of endothelial cells. Exposure to myocyte exosomes downregulated miR-320 target genes, such as Hsp20, IGF-1, and Ets2. This is one mechanism of diabetes-induced myocardial deficiency [81]. The levels of pro-angiogenic miR-126 were also decreased under hyperglycemic conditions in the same study [81]. miR-126 is transported in circulating microvesicles (cMVs) and endothelial progenitor cell-derived MVs (EPC-MVs). Endothelial progenitor cell (EPC)-derived MVs modulate proliferation and influence endothelial cell survival [83]. It has been demonstrated that miR-126 modulates the function of endothelial progenitor cells by targeting the VEGFR2-associated signal transduction. Under conditions of hyperglycemia, the levels of miR-126 and VEGFR2 are decreased. In such conditions, EPCs experience oxidative stress, which leads to apoptosis. Additionally, their migration ability and functions in T2DM patients are suboptimal [84]. Among other miRNAs that affect endothelial cell function, miR-200B is worth mentioning. It is responsible for inhibiting vascular endothelial cell proliferation by downregulating the expression of GATA2, the Ets-1 gene, vascular endothelial growth factor (VEGF), and vascular endothelial growth factor receptor 2 (VEGFR-2) [85].

### 5.3. Inflammation and Fibrosis

Chronic inflammation and myocardial fibrosis occur in cardiac diseases caused by diabetes mellitus (Figure 3) [86,87].

One of the factors deserving attention as a cause of the pathological process of inflammation is miR-155. An in vivo study confirmed the increased expression of miR-155 in high-glucose-treated cells [88]. Its adverse effects are manifested at the cellular level by stimulating fibroblasts to secrete collagen and at the gene level by silencing the anti-inflammatory gene. Macrophage-derived exosomes containing this particular miRNA affect cardiac fibroblasts and influence cardiac injury by diminishing the expression of gene Suppressor of Cytokine Signaling 1 (Socs1) and elevating the inflammation of the heart muscle [89,90]. The Socs1 gene, by inhibiting Jak kinase activity, is included in a family of anti-inflammatory proteins and a unit of a negative feedback system [91,92]. The overexpression of mir-155 causes a decreased Socs1 level and the accretion of cardiac fibroblast ignition, resulting in the secretion of collagen and actin [89]. Furthermore, macrophage-derived exosomes and inflammation also stimulate fibroblasts to excrete pro-inflammatory cytokines such as IL-1B, IL-6, and TNF-a [89,93]. Further increased cardiac inflammation results in the enhanced secretion of pro-inflammatory cytokines, which causes a continuity of the inflammatory process. The aforementioned cytokines, through the activation of fibroblasts, contribute to the fibrosis process [94].

Cardiac fibrosis progression is caused by the increased level of collagen I and alpha-smooth muscle actin [95,96]. Previously mentioned miR-155 causes the overexpression of both of these proteins and undermines the Nrf2/HO-1 signaling pathway responsible for preventing excessive extracellular matrix accumulation [88]. Taking into account these mechanisms as well as the augmentation of TNF-alpha, which also induces fibrosis, miR-155 contributes to fibrosis induction [97]. Another stimulus factor of fibrotic remodeling is human antigen R (HuR)—an mRNA-stabilizing protein. Under diabetic conditions, it is transferred via macrophage-derived exosomes into fibroblasts, which stimulate inflammation and fibrosis, resulting in cardiac dysfunction [98]. An increased level of HuR protein results in cardiac fibrosis induced by angiotensin-II, which is a promoting factor of cardiomyocyte hypertrophy and cardiac fibroblast proliferation [98,99].

### 5.4. Calcium Dyshomeostasis

Disrupted Ca^2+^ cycling, noticeable in the early stages of DCM in cardiomyocytes, leads to contractile and diastolic dysfunction (Figure 3) [86,87]. Bone marrow-derived macrophage-derived extracellular vesicles contain miR-25 in cardiac damage conditions [100]. In vitro and in vivo studies have shown that miR-25 is upregulated in heart failure. This miRNA is an important suppressor of SERCA2a [101]. SERCA2a is the main machinery responsible for Ca^2+^ uptake during excitation, leading to contraction. Therefore, miR-25 impairs intracellular calcium handling in cardiomyocytes by delaying SERCA2a-dependent calcium uptake [101].

MiR-155, by exacerbating inflammation and increasing the level of IL-1B and TNF-a, results in Ca^2+^ dyshomeostasis. IL-1beta decreases the expression of calcium-regulating genes, among them being SERCA2, the calcium release channel (CRC), and the voltage-dependent calcium channel (VDCC) [102]. Moreover, TNF-alpha triggers contractile dysfunction through two mechanisms: early and delayed. The early pathway is partially mediated by sphingosine, which inhibits the sarcoplasmic ryanodine receptor that lessens Ca^2+^ release [103]. On the other hand, the delayed effect is mediated via stimulating the production of nitric oxide synthase and therefore promoting NO synthesis [104,105]. Through the cGMP-dependent blockage of L-type Ca^2+^ channels, NO prevents calcium influx and desensitizes the myofilaments to Ca^2+^ [103,105]. 

### 5.5. Senescence

When discussing diabetes, one cannot overlook the phenomenon of cellular aging. In type II diabetes, metabolic changes associated with hyperglycemia lead to the accumulation of senescent cells in multiple organs [106]. Senescent cells are characterized by irreversible cell cycle arrest, accompanied by a decline in cellular function. For instance, hyperglycemic conditions result in the aging of cardiac stem cells, which leads to the pathological remodeling and dysfunction of cardiac tissue [106]. Recent studies have shown an increase in the number of released EVs from senescent cells and a change in their cargo profile [107]. The senescence-associated secretory phenotype (SASP) involves the release of pro-inflammatory cytokines and chemokines by senescent cells. [108]. Extracellular vesicles, soluble factors, and metalloproteinases serve as means of communication among senescent cells within their microenvironment. The signaling pathways precisely activated by extracellular vesicles in senescence remain incompletely understood [109]. However, the induction of inhibitors of nuclear factor-κB (IκB) kinases (IKK) α, β, and ε via EVs resulting in NF-κB activation was described [109]. As previously mentioned, chronic inflammation contributes to the development of diabetes. It is not entirely clear whether cellular aging is a result of chronic inflammation or a source of chronic inflammation. However, pro-inflammatory factors such as IL-1, IL-8, and IL-6 are secreted in the SASP [110]. Extracellular vesicles derived from healthy cells may play a role in modulating the aging process through various mechanisms, including the transmission of microRNAs, the secretion of antioxidant enzymes, and metabolic alterations. It has been proposed that extracellular vesicles can influence the SASP by reducing pro-inflammatory changes in the microenvironment, which are caused by the accumulation of senescent cells [107].

## 6. Beneficial Effects of Extracellular Vesicles in Diabetic Cardiomyopathy

EVs, due to their indicator and cardioprotective characteristics, may become an important tool in the diagnosis and therapy of cardiovascular disorders, offering alternative curative options [111]. A summary of the EVs’ beneficial effects in DCM is presented in Table 2.

The EVs released during oxidative stress contain antioxidant molecules that modulate the oxidative stress response, thereby protecting target cells from damage [127]. Understanding the molecular processes that regulate EV release is critical for developing treatment options for oxidative stress-related diseases, which rely on either the elimination of damaging compounds or the transport of antioxidants via EVs [128]. Extracellular vesicles contain a molecular cargo associated with oxidative stress and therefore indicate the redox status of a cell or tissue [128]. Thus, EVs can be used as a biomarker to monitor the progression of oxidative stress-related diseases [128]. Wang et al. observed a reduction in the expression of the Hsp 20 protein in response to both acute and chronic hyperglycemia in their study on mice [117]. Hsp 20 is one of the factors involved in the development of diabetic cardiomyopathy. Tsg101 is an upstream factor of the exosome biogenesis pathway that directly interacts with Hsp 20 to activate the exosome production signaling pathway. The study conclusively demonstrates that exosomes released from cardiomyocytes, modified by Hsp 20 heat shock proteins, have a protective effect through autocrine and paracrine action [128]. The released exosomes lead to reduced apoptosis and increased myocardial blood vessel density. Importantly, exosome production in cardiomyocytes increased with the elevation of Hsp 20 levels. An increase in Hsp 20 levels in cardiomyocytes protects against cardiomyopathy [128]. In vitro studies on cardiomyocytes subjected to oxidative stress demonstrate the anti-apoptotic properties of miR-25 by preserving Bcl-2 expression [112]. Bcl-2 protein belongs to the group of inhibitors of apoptosis [129]. Through protecting Bcl-2 expression, miR-25 inhibits the programmed death pathway [112]. It is commonly recognized that oxidative stress and cardiomyopathy are closely related to decreased levels of miR-499, miR-133a, and miR-133b [113,114,115]. The lower expression of these miRNAs has been reported in DM cardiomyocytes [130]. Also, medicines can affect the secretion of EVs by different cells, and because of that, we can study the effects of drugs on their composition and therapeutic abilities. For example, ticagrelor pre-treatment of cardiomyocytes can change the profile of extracellular vesicles derived from them and suppress cellular stress in hyperglycemic cardiomyocytes and their apoptosis via secreting changed EVs. The changing profile of EVs increases the expression level of miR-133a, miR-133b, and miR-499 [116].

Excessive insulin resistance mediates endothelial dysfunction and can lead to the death of cardiomyocytes through the activation of the RAS system [131]. The study showed that muscle-derived exosomal miR-133b improves insulin sensitivity by downregulating forkhead class O1 (FoxO1) transcription factors in mice [118]. The release of exosomes (containing miR-133b) from muscles occurs as a result of high-intensity interval training (HIIT) [118]. Such exosomes could be useful in the treatment of metabolic disorders leading to diabetic cardiomyopathy. Other miRNAs are also involved in improving insulin sensitivity, e.g., exosomal miR-690 targets NAD kinase (Nadk), which is involved in insulin signaling and the modulation of macrophage inflammation [119]. Another beneficial molecule is miR-145, which decreases Ca^2+^/calmodulin-dependent protein kinase II (CaMKII) [120]. CaMKII stimulates several ROS-induced apoptotic pathways in the heart via increasing ROS production. Therefore, miR-145 by the inhibition of CaMKII reduces ROS production and prevents [Ca^2+^] increases [94,120]. miR-145 was found in exosomes derived from bone marrow-derived mesenchymal stem cells and in exosomes from adipose-derived stem cells. Still, their presence and impact on the development of cardiomyopathy require further studies [121,122]. Human mesenchymal stem cell (hMSC)-derived exosomes are the cause of increases in human-engineered cardiac tissue contractility and related calcium handling [123]. The main cardioactive hMSC exosomal miRNA is miR-21-5p due to its impact on the increase in calcium handling gene expression such as mRNA of LTCC or SERCA2a. Moreover, this particular miRNA affects the PI3K signal cascade thereby increasing contractility [124]. In vitro and in vivo studies have demonstrated that extracellular vesicles (EVs), released from activated endothelial cells (ECs), contain miRNA-222 which presents anti-inflammatory effects on adhesive ECs by targeting and reducing ICAM-1 expression [125]. However, miRNA-222 expression is reduced in EVs generated under hyperglycemic conditions and results in decreased anti-inflammatory properties [125]. Mesenchymal stem cell-derived exosomes decrease the level of TGF-beta1 and Smad2 mRNAs and proteins [126]. Type I and III collagen fiber production is induced by these pro-inflammatory cytokines [132]. In vitro studies have shown that under hyperglycemic conditions, TGF-beta mRNA and protein are upregulated. MSC-derived exosomes, by downregulating the expression of the TGF-beta gene decrease the level of collagen in the heart and exert a positive effect on cardiac fibroblasts [126]. The same TGF-beta pathway is mediated by exosomes derived from cardiac microvascular epithelial cells [133].

## 7. Conclusions

The role of EVs in the pathobiology of DCM is a still-evolving topic. As mentioned above, the expression of several molecules differs between healthy and diabetic individuals. The more thorough the research concerning the gradation of EVs’ presence during the development of DCM is, the easier the task of deciding on the therapy introduction is. EVs have great indicatory potential since they are easily obtainable from almost any body fluid such as blood, urine, and saliva [134,135,136]. Taking into consideration the potential therapeutic benefits of sodium-glucose co-transporter 2 inhibitors (SGLT2i) on cardiac remodeling in addition to their blood sugar-lowering action, the exploration of the markers at each stage of the disease is crucial for proper treatment [137]. A better understanding of the fluctuation of the biomarkers can ensure the introduction of therapy before irreversible heart muscle lesions. Several clinical trials are currently investigating EVs’ diagnostic potential (Table 3). For instance, the characterization of different non-coding RNAs in the blood and saliva of HF patients is of interest (NCT06169540, NCT03268135, NCT05726695). Moreover, EVs could be a great advancement in therapy monitoring, e.g., after bariatric surgery; thus, such clinical trials are also currently being conducted (NCT06401876, NCT06408961).

The exploration of the topic of EVs in DCM shows their involvement in the pathophysiology to be a double-edged sword. On the one hand, they detrimentally influence cells causing the acceleration of the pathology. As mentioned above, the cargo of EVs may adversely act on several levels. Reducing the effects of EVs may include several actions: downregulating the production of EVs’ cargo, the inhibition of the loading process, decreasing the EVs’ release, and blockage of the uptake by target cells. Hopefully, further studies will find proper methods to selectively act on these mechanisms, thus reducing EVs’ deleterious activity. On the other hand, several positive effects of EVs were described in the literature. This knowledge is crucial for the invention of new therapeutic options. It is plausible that with better-obtained methods and purification processes, EVs might become the DCM drug itself. Of note, currently, an extracellular vesicle-enriched secretome of cardiovascular progenitor cells is being investigated as a treatment option for patients with severe symptoms of left ventricle dysfunction in non-ischemic dilated cardiomyopathy (NCT05774509). Mesenchymal stem cell (MSC)-derived EVs are also extensively investigated as a treatment option in lung injury diseases such as pneumonia [138,139]. In mouse models of P. aeruginosa-induced lung injury, the nebulization of EVs from MSCs increases the survival rate by decreasing lung inflammation and histological severity [138]. Additionally, they do not cause severe adverse effects when administered via nebulization in healthy individuals [138]. Similarly, the good tolerance of EV nebulization was shown in seven mild COVID-19 pneumonia cases, where the therapy induced quicker pulmonary lesion absorption [139]. Wound healing is another field of research where the EVs’ potential is being investigated. The use of adipose tissue stem cell-derived EVs decreased downtime and increased improvement after fractional CO_2_ laser sessions [140]. The use of EVs in diabetic foot ulcers is currently being investigated (NCT06319287).

It is noteworthy that treatments with several drugs used in cardiology alter EVs’ cargo. Drugs such as ticagrelor can positively alter the composition of EVs [130]. With some further insight, the therapy selection can take into consideration additional aspects such as the modulation of EVs as a part of the therapeutic potential of the drug. Intriguingly, one clinical trial proposed the improvement in heart function due to levosimendan treatment to be partially caused by the alterations in the levels of three miRNAs in EVs: miR-660-3p, miR-665, and miR-1285-3p (NCT04950569). However, the results of the study remain unobtainable.

With extensive research regarding extracellular vesicles, controversies surrounding the topic have arisen. To date, there are no standardized methods of collection, purification, and clarification of the molecules, which makes the comparison between studies considering EVs unfeasible. Additionally, the majority of the studies in the field are preclinical. We still await good-quality clinical research, which would elucidate whether the application of EVs in therapy and diagnosis is possible. Preclinical research is the first step in the long process of the introduction of EV technology in therapy. Another problem is the production technology of EVs. For now, it seems unlikely that the production process of EVs will be described shortly because even the cellular mechanism of secretion of EVs and the choice of the cellular target is still being investigated. We are still at the beginning of the way, but the future is promising.

To sum up, the manipulation of EV levels and cargo could become a useful tool for alterations in the intercellular communication between cells in the cardiac microenvironment; thus, EVs could become a huge ally in DCM treatment.

## Figures and Tables

**Figure 1 ijms-25-06117-f001:**
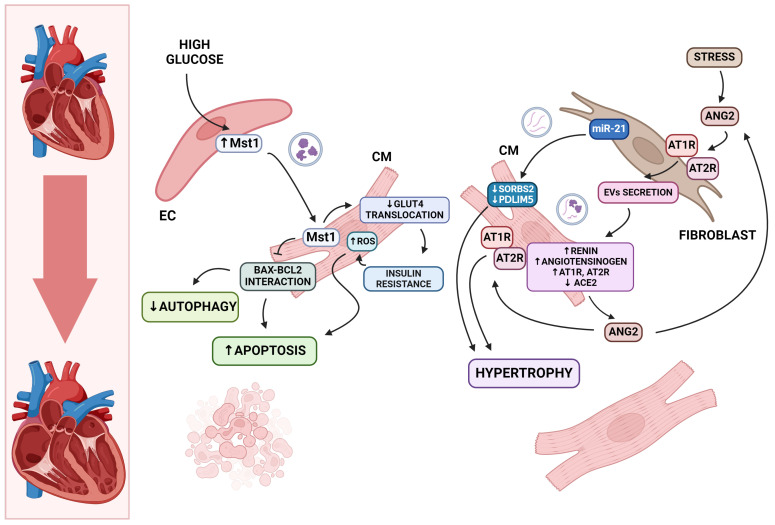
Diabetes induces the death of cardiomyocytes. This is due to the secretion of Mst1-containing exosomes by hyperglycemic-conditioned endothelial cells, which are internalized by cardiomyocytes. The uptake of Mst1 leads to an increase in proapoptotic Bax levels and decreased expression of antiapoptotic Bcl-2 protein. Additionally, GLUT4 translocation is also inhibited by Mst1, leading to insulin resistance and increased reactive oxygen species, which further elevate cellular stress, thus inducing apoptosis. The weakening of the heart muscle is compensated by the hypertrophy of cardiomyocyte remnants. One of the molecules included in this process is miR-21, which downregulates PDLIM5 and SORBS2 under high-glucose conditions. Additionally, cellular stress can also lead to cardiac muscle hypertrophy. In such conditions, the RAS system is stimulated, promoting EVs’ secretion by fibroblasts. These molecules interact with cardiomyocytes, promoting renin, angiotensin, and angiotensin receptors’ expression. As a result, the levels of angiotensin II in the cardiac microenvironment are elevated, which further stimulates fibroblast EVs’ release and additionally interacts with upregulated angiotensin receptors on the cardiomyocytes, leading to hypertrophy. These alterations lead to structural changes in the heart muscle. Abbreviations: AT1R—angiotensin II type 1 receptor; AT2R—angiotensin II type 2 receptor; ACE 2—angiotensin-converting enzyme 2; ANG 2—angiotensin II; GLUT4—glucose transporter type 4; Mst1—mammalian sterile 20-like kinase 1; EC—endothelial cell; CM—cardiomyocyte; ROS—reactive oxygen species; SORBS2—Sorbin And SH3 Domain Containing 2; PDLIM5—PDZ And LIM Domain 5; miR—microRNA; EVs—extracellular vesicles. Created in biorender.com (accessed on 6 January 2024).

**Figure 2 ijms-25-06117-f002:**
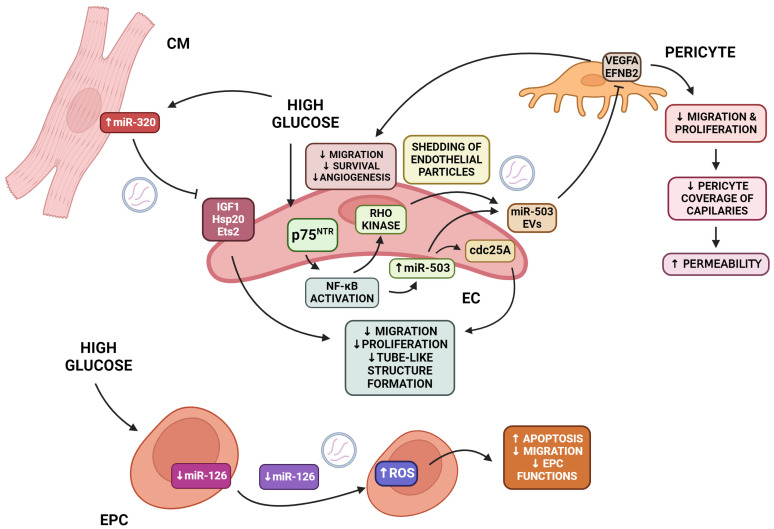
Hyperglycemia affects endothelial homeostasis via extracellular vesicles. Hyperglycemia leads to the increased release of cardiomyocyte-derived EVs containing miR-320-inhibiting functions of several pro-angiogenic proteins within endothelial cells. Additionally, elevated blood sugar levels influence endothelial cells, leading to an increase in miR-503 expression, which impacts angiogenesis by downregulating cdc25A, but also may be transmitted via EVs to pericytes, decreasing their VEGFA and EFNB2 levels. This action suppresses the migration and proliferation of both ECs and pericytes, which translates to increased vessel permeability and decreased angiogenesis. Lastly, high glucose results in the downregulated transcription and release of pro-angiogenic miR-126 from EPC, thus increasing EPC apoptosis and inhibiting their positive influence on ECs. Abbreviations: CDC25A—cell division cycle 25 A; CM—cardiomyocyte; EC—endothelial cell; EFNB2—Ephrin B2; EPC—endothelial progenitor cells; Ets2—transcription factor; Hsp 20—heat shock protein 20; IGF—insulin-like growth factor; MiR—microRNA; NF-KB—nuclear factor kappa B; P75NTR—p75 neurotrophin receptor; ROS—reactive oxygen species; VEGFA—vascular endothelial growth factor A. Created in biorender.com (accessed on 6 January 2024).

**Figure 3 ijms-25-06117-f003:**
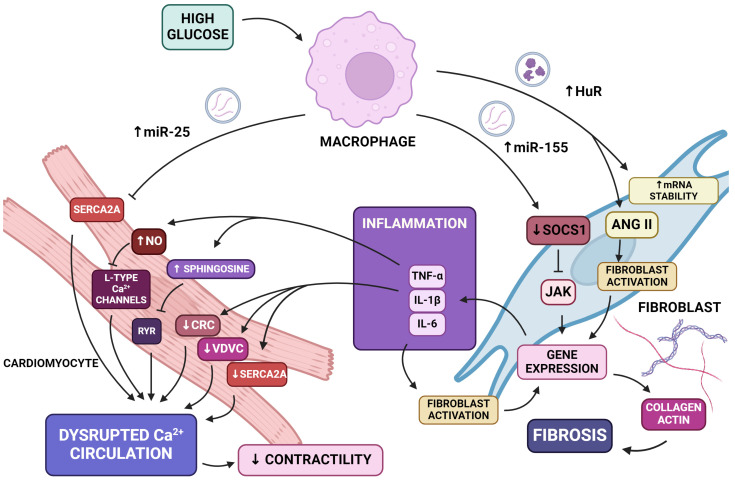
Effects of extracellular vesicles and their contents on the function of cardiac fibroblasts and cardiomyocytes under hyperglycemic conditions. Exosomes secreted in diabetes can contribute to calcium dysregulation through reduced gene expression or the blocking of ion transport channels; exosomes can enhance inflammatory processes and fibrosis in cardiac muscle tissue through increased gene expression and the stimulation of fibroblasts. The aforementioned processes are components of the pathophysiological process leading to diabetic cardiomyopathy. Abbreviations: Ang II—angiotensin II; CRC—calcium release channel; HuR—human antigen R, JAK—Janus kinase; IL—interleukin; miR—micro RNA; NO—nitric oxide; RYR—ryanodine receptor; Socs1—Suppressor of Cytokine Signaling 1, TNF-α—tumor necrosis factor-alpha; VDCC—voltage-dependent calcium channel; SERCA2a—sarcoplasmic/endoplasmic reticulum Ca^2+^-ATPase. Created in biorender.com (accessed on 6 January 2024).

**Table 1 ijms-25-06117-t001:** Nomenclature and classification of Extracellular Vesicles.

Nomenclature of Extracellular Vesicles (EVs)			References
Classical (based on biogenesis) *	exosomes	exocytosis of MVB	[27,28,29,30,31,32]
	microvesicles	direct budding from PM	
	apoptotic EVs	apoptosis	
Size	small	<200 nm	
	medium/large	>200 nm	
Density	low	each range should be defined, e.g., exosome and microvesicle density—1.08–1.19 g/mL; apoptotic EV density—1.16–1.28 g/ml	
	medium		
	high		
Biochemical composition	e.g., CD63, CD81, CD9-positive EVs—classical exosomes; annexin-V-positive EVs—apoptotic EVs; ARRDC1-positive EVs—ARMM		
Cellular origin	e.g., myocardial EV, endothelial EV, oncosome, fibroblast-derived EV		
Condition	e.g., hypoxic EV, apoptotic EV		

* classification not recommended by International Society for Extracellular Vesicles; abbreviations: ARMM—ARRDC1-mediated microvesicles; ARRDC1—arrestin-domain-containing protein 1; CD—cluster of differentiation; EV—extracellular vesicle; MVB—multivesicular body; PM—plasma membrane.

**Table 2 ijms-25-06117-t002:** Beneficial effects of extracellular vesicles in diabetic cardiomyopathy.

**Type of miRNA**	**Type of Extracellular Vesicles**	**Secreting Cells**	**Target Cells**	**Mechanism of Changes**	**Ref.**
miR-25	No data	No data	Cardiomyocytes	Anti-apoptotic effect	[112]
miR-133a/b	Extracellular Vesicles	Cardiomyocytes	Cardiomyocytes	Anti-apoptotic effect	[113,114,115,116]
No data	Exosomes	Cardiomyocytes	Cardiomyocytes	The anti-apoptotic effect, increased myocardial vascular density	[117]
miR-133b	Exosomes	Muscle	Hepatocytes	Improved insulin sensitivity	[118]
miR-690	Exosomes	Macrophages	Adipocytes, hepatocytes	Improved insulin sensitivity	[119]
miR-145	Exosomes	Bone marrow mesenchymal stem cells and adipose-derived stem cells	Cardiomyocytes	Regulation of intracellular calcium level	[94,120,121,122]
miR-21-5p	Exosomes	Human mesenchymal stem cells	Cardiomyocytes	Regulation of intracellular calcium level and contractile improvement	[123,124]
miR-222	Extracellular Vesicles	Endothelial cells	Endothelial cells	Anti-inflammatory effect	[125]
No data	Exosomes	Mesenchymal stem cells	Cardiac fibroblasts	Inhibition of fibrosis	[126]

**Table 3 ijms-25-06117-t003:** Clinical trials indicating future directions of EV research in DCM.

Trial Number	Status	Disease(s)	Interventional or Observational	Aim of the Study	Potential Application in DCM Research
NCT05774509	recruiting	non-ischemic dilated cardiomyopathy	interventional	assessment of the safety and efficacy of the extracellular vesicle-enriched secretome of cardiovascular progenitor cells in severely symptomatic patients with non-ischemic dilated cardiomyopathy	therapy
NCT06169540	recruiting	acute decompensated heart failure and chronic heart failure	observational	determining the relationship between the levels of RNA, including ones in extracellular vesicles from different organs in the blood and the saliva of patients with HF	diagnosis
NCT03268135	recruiting	heart failure, aortic stenosis	observational	investigating global transcriptome to determine the expression profile of different RNAs in patients with heart failure and aortic stenosis	diagnosis
NCT05726695	active, not recruiting	heart failure with reduced ejection fraction	observational	determining the analytical characteristics of the microRNA enzymatic immunoassay method and various relations among miRNA biomarkers and heart failure	diagnosis
NCT04950569	unknown	heart failure	interventional	determining the relationships between several clinical and laboratory findings (including myocardial miRNAs) after levosimendan treatment	therapy
NCT02138331	unknown	T1DM	interventional	assessing the relationship between intravenous infusion of microvesicles from mesenchymal stem cells and β-cell mass as well as the glycemic control in T1DM patients	therapy
NCT02459106	active, not recruiting	T2DM	observational	investigation of effects of fat tissue-released miRNA on biology and insulin sensitivity of skeletal muscle	diagnosis/therapy
NCT05139914	recruiting	T2DM	interventional	assessment of the impact of dapagliflozin on endothelial cell health (including non-coding RNA assessment)	therapy, therapy monitoring
NCT05259449	recruiting	T2DM	interventional	determination of the efficacy of nutritional and physical education on health-related variables (such as exosome profile) in T2DM patients	therapy monitoring
NCT06401876	not yet recruiting	T2DM, obesity	observational	profiling the cargo of extracellular vesicles of obese diabetic patients before and after bariatric surgery	therapy monitoring
NCT06408961	recruiting	obesity, cardiometabolic disease	observational	researching the content and function of extracellular vesicles in obese patients receiving bariatric surgery	therapy monitoring

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
