# Peer review of "Extracellular Vesicles in Diabetic Cardiomyopathy—State of the Art and Future Perspectives"

_ijms, 2024, doi:10.3390/ijms25116117_

Round 1

Reviewer 1 Report

Comments and Suggestions for Authors

The review by Zygmunciak and colleagues provides an intriguing analysis of recent investigations into the potential involvement of extracellular vesicles (EVs) in diabetic cardiomyopathy (DCM), suggesting their utility as a treatment tool for diabetic patients. 

The authors extensively examine the detrimental impact of diabetes on cardiac tissue outcomes. However, given that the adverse effects on cardiac function and tissue architecture in diabetes have been linked to cellular senescence and the expression of senescence markers such as CDKIs p16, p21, and b-gal, as well as mediators and constituents of the senescence-associated secretory phenotype (SASP) (please see doi:10.2337/db21-0536; doi:10.3390/ijms24021132; doi:10.3389/fendo.2022.869414 doi:10.1152/ajpheart.00287.2018; doi: 10.3390/antiox11020208; https://academic.oup.com/endo/article/162/10/bqab136/6345039), it could be worthwhile to also consider discussing these damaging events. Moreover, it would be interesting to analyze the potential of EVs in alleviating cellular senescence (please see DOI: 10.2174/1570161116666180820115726 ; doi: 10.3390/ijms232314632; doi: 10.3390/ijms21103725). 

Author Response

Warsaw, 16th of May 2024

Dear Reviewers,

We are very grateful for your careful review of our work. We addressed all of the issues raised in your comments and provided the requested additional data in the revised version of the manuscript. All the changes in regard to the previous version of the paper were highlighted in the text. We hope you find the revised manuscript suitable for publication in International Journal of Molecular Sciences.

Sincerely yours,

Authors

Reviewer 1

The review by Zygmunciak and colleagues provides an intriguing analysis of recent investigations into the potential involvement of extracellular vesicles (EVs) in diabetic cardiomyopathy (DCM), suggesting their utility as a treatment tool for diabetic patients. 

The authors extensively examine the detrimental impact of diabetes on cardiac tissue outcomes. However, given that the adverse effects on cardiac function and tissue architecture in diabetes have been linked to cellular senescence and the expression of senescence markers such as CDKIs p16, p21, and b-gal, as well as mediators and constituents of the senescence-associated secretory phenotype (SASP) (please see doi:10.2337/db21-0536; doi:10.3390/ijms24021132; doi:10.3389/fendo.2022.869414 doi:10.1152/ajpheart.00287.2018; doi: 10.3390/antiox11020208; https://academic.oup.com/endo/article/162/10/bqab136/6345039), it could be worthwhile to also consider discussing these damaging events. Moreover, it would be interesting to analyze the potential of EVs in alleviating cellular senescence (please see DOI: 10.2174/1570161116666180820115726 ; doi: 10.3390/ijms232314632: doi: 10.3390/ijms21103725). 

Thank you very much for the kind words and comments regarding our paper. Indeed, the senescence plays a huge role in diabetic cardiomyopathy. We have carefully read all of the linked papers and conducted a thorough research concerning aging processes, extracellular vesicles and cardiomyopathy. However, the link between extracellular vesicles and cellular senescence in diabetic cardiomyopathy is still not thoroughly studied and we couldn’t find papers clearly linking all of the topics. Considering all of the above, we decided not to include senescence in our publication, but we hope to extend our knowledge in the field, which may result in a separate publication in the near future.

Reviewer 2 Report

Comments and Suggestions for Authors

Authors report a review on the diabetic cardiomyopathy and in particular extracellular vescicles. The review is well conducted and the results are clear and useful for the reader. I have only few suggestions. 

In the manuscript shou be clear if it is referred to DM I/ DM II or other forms of diabetes. 

A section Materials and methods with the methods of the literature research and selection should be added

Author Response

Warsaw, 16th of May 2024

Dear Reviewers,

We are very grateful for your careful review of our work. We addressed all of the issues raised in your comments and provided the requested additional data in the revised version of the manuscript. All the changes in regard to the previous version of the paper were highlighted in the text. We hope you find the revised manuscript suitable for publication in International Journal of Molecular Sciences.

Sincerely yours,

Authors

Reviewer 2

Authors report a review on the diabetic cardiomyopathy and in particular extracellular vescicles. The review is well conducted and the results are clear and useful for the reader. I have only few suggestions. 

In the manuscript shou be clear if it is referred to DM I/ DM II or other forms of diabetes. 

A section Materials and methods with the methods of the literature research and selection should be added

Thank you abundantly for your review and most useful suggestions. We have updated the manuscript accordingly. We haven’t specified the type of the disease, since our review talks about both type 1 and 2 diabetes mellitus. Our work includes mostly preclinical studies. We have added the information regarding type of the DM, when provided. Additionally, we have added a new section called “Materials and methods” including our search criteria.

Reviewer 3 Report

Comments and Suggestions for Authors

Authors reviewed the interesting role of EV son DCM, highlighting the mechanism of secretion and action under different cardiac responses. The work could be improved if authors:

-              Reduce or even delete the section 3, since DCM pathology has been largely described.

-              Fig1. The apoptotic response as scheme may not be understood. Please, change the picture

-              EVs may cargo other than nucleic acid or proinflammatory molecules. Whole organelles such as mitochondria have been described in other pathologies (i.e., cardio renal syndrome) and may also occurs in DCM complications. Organelles may be transferred among cardiac cells or among different tissues. Please, discuss.

Author Response

Warsaw, 16th of May 2024

Dear Reviewers,

We are very grateful for your careful review of our work. We addressed all of the issues raised in your comments and provided the requested additional data in the revised version of the manuscript. All the changes in regard to the previous version of the paper were highlighted in the text. We hope you find the revised manuscript suitable for publication in International Journal of Molecular Sciences.

Sincerely yours,

Authors

Reviewer 3

Authors reviewed the interesting role of EV son DCM, highlighting the mechanism of secretion and action under different cardiac responses. The work could be improved if authors:

-              Reduce or even delete the section 3, since DCM pathology has been largely described.

-              Fig1. The apoptotic response as scheme may not be understood. Please, change the picture

-              EVs may cargo other than nucleic acid or proinflammatory molecules. Whole organelles such as mitochondria have been described in other pathologies (i.e., cardio renal syndrome) and may also occurs in DCM complications. Organelles may be transferred among cardiac cells or among different tissues. Please, discuss.

Thank you very much for a kind review of our work. We believed that the information provided in the Diabetic Cardiomyopathy section of the article are essential for the understanding of the following parts of the articles regarding EVs.

Thank you for drawing our attention to the unclarity of the 1 Figure. Instead of changing it, we decided to provide better description of the processes that take place in cardiomyocytes. You can find the corrections in the description box under the Figure.

It is true that cellular organelles could be transferred via extracellular vesicles, mostly by apoptotic bodies. However, after a thorough search in Medline we couldn’t find a single article describing such action regarding diabetic cardiomyopathy. We hope that the future research would elucidated this particular pathophysiological process.

Reviewer 4 Report

Comments and Suggestions for Authors

Comments:

Strengths of the article Comprehensive Coverage:

The EVs' intricacy in the pathophysiology of DCM is clearly demonstrated by the article, which further elaborates on the deleterious and beneficial sides. The current work is a highlight in its strength for developing a clear molecular understanding of how high glucose levels, in concert with associated miRNAs, contribute toward the development of DCM. Clinical Significance

In this regard, diagnostic and therapeutic development center around EVs, which find high relevance in current needs for the management of chronic conditions such as DCM. In particular, mention of practical, real-world application, such as taking body fluids for EV extraction, bridges basic research to clinical practice. Areas of Improvement Mechanistic Details

Although it defines which miRNAs are involved and what the impact of high glucose on EVs is, it could be improved by much more in-depth information about the mechanisms. For example, in-depth information about signaling cascades and how EVs modify the specified cascades would be more informative to the reader and furnish a comprehensive view of potential points of intervention. Evidence-Based Data:

The article keeps quoting the findings from the studies, but the actual data or the context of the experiments is missing. It would have been more desirable that such important studies were specifically summarized in terms of sample sizes, designs of studies, and the degree of statistical significance, in support of the conclusions made. Conflicting Views and Controversies

The dual role of EVs in DCM is intriguingly debated. However, the present article could be of more benefit if it applies a balanced view in consideration of contrasting findings or discusses the controversies in the area. For instance, beneficial effects of EVs have not been replicated. Future Directions:

The paper shortly mentioned, among other things, the future needs of this research, but not the specific challenges and opportunities of EV research in regard to DCM. It deserves a paragraph listing the research directions that can be followed, e.g., the development of new EV-based biomarkers or therapeutic agents. Clinical Application:

Although the article talks about the therapeutic potential of EVs, it does not really allude to a plainly visible pathway for clinical translation. Thus, a discussion of current barriers to clinical application that may include EV purification, scalability of production, or regulatory challenges could render the discussion more useful to clinicians and researchers.

Ideas:

Add More Experimental Findings:

More data on experiments, case studies, and meta-analyses should be included in this article to enhance the credibility that EVs have therapeutic and diagnostic roles in DCM.

Touch on Clinical Implications: Future directions should include a more detailed discussion of how these findings might be translated into the clinical setting, either from the perspective of possible clinical trials, regulatory consideration by the FDA, or integration within existing treatment protocols. This multidisciplinary Here, interdisciplinary research in cardiology, endocrinology, and molecular biology may open up a novel avenue with the solution of DCM by EVs. Describing Limitations It should also include a critical appraisal of the limitations within the current understanding of EV roles in DCM. This then allows open discussion to set future research and clinical expectation scopes.

Author Response

Warsaw, 16th of May 2024

Dear Reviewers,

We are very grateful for your careful review of our work. We addressed all of the issues raised in your comments and provided the requested additional data in the revised version of the manuscript. All the changes in regard to the previous version of the paper were highlighted in the text. We hope you find the revised manuscript suitable for publication in International Journal of Molecular Sciences.

Sincerely yours,

Authors

Reviewer 4

Strengths of the article Comprehensive Coverage:

The EVs' intricacy in the pathophysiology of DCM is clearly demonstrated by the article, which further elaborates on the deleterious and beneficial sides. The current work is a highlight in its strength for developing a clear molecular understanding of how high glucose levels, in concert with associated miRNAs, contribute toward the development of DCM.

Clinical Significance

In this regard, diagnostic and therapeutic development center around EVs, which find high relevance in current needs for the management of chronic conditions such as DCM. In particular, mention of practical, real-world application, such as taking body fluids for EV extraction, bridges basic research to clinical practice.

Areas of Improvement Mechanistic Details

Although it defines which miRNAs are involved and what the impact of high glucose on EVs is, it could be improved by much more in-depth information about the mechanisms. For example, in-depth information about signaling cascades and how EVs modify the specified cascades would be more informative to the reader and furnish a comprehensive view of potential points of intervention.

Evidence-Based Data:

The article keeps quoting the findings from the studies, but the actual data or the context of the experiments is missing. It would have been more desirable that such important studies were specifically summarized in terms of sample sizes, designs of studies, and the degree of statistical significance, in support of the conclusions made.

 Conflicting Views and Controversies

The dual role of EVs in DCM is intriguingly debated. However, the present article could be of more benefit if it applies a balanced view in consideration of contrasting findings or discusses the controversies in the area. For instance, beneficial effects of EVs have not been replicated.

Future Directions:

The paper shortly mentioned, among other things, the future needs of this research, but not the specific challenges and opportunities of EV research in regard to DCM. It deserves a paragraph listing the research directions that can be followed, e.g., the development of new EV-based biomarkers or therapeutic agents.

Clinical Application:

Although the article talks about the therapeutic potential of EVs, it does not really allude to a plainly visible pathway for clinical translation. Thus, a discussion of current barriers to clinical application that may include EV purification, scalability of production, or regulatory challenges could render the discussion more useful to clinicians and researchers.

Ideas:

Add More Experimental Findings:

More data on experiments, case studies, and meta-analyses should be included in this article to enhance the credibility that EVs have therapeutic and diagnostic roles in DCM.

Touch on Clinical Implications: Future directions should include a more detailed discussion of how these findings might be translated into the clinical setting, either from the perspective of possible clinical trials, regulatory consideration by the FDA, or integration within existing treatment protocols. This multidisciplinary Here, interdisciplinary research in cardiology, endocrinology, and molecular biology may open up a novel avenue with the solution of DCM by EVs. Describing Limitations It should also include a critical appraisal of the limitations within the current understanding of EV roles in DCM. This then allows open discussion to set future research and clinical expectation scopes.

Thank you very much for the review and accurate comments on our paper. We find your insight extremely helpful for the setting the future clinical research perspective regarding extracellular vesicles in diabetic cardiomyopathy. We would have included more meta-analyses, systematic reviews, or clinical studies, however, such articles are still lacking. Therefore, we used almost solely the data from preclinical analyses. First meta-analyses on extracellular vesicles in more common diseases have been recently published. Taking under consideration, that diabetic cardiomyopathy is still being discussed as an individual disease and the research regarding this condition is limited, we couldn’t find more data on the matter. We believe that such information will be available soon with the ongoing collective effort of the scientific community.

When it comes to the description of molecular pathways, we believe that we provided enough information for the understanding of the problem in a straightforward manner. The inclusion of more detailed mechanisms of described processes would decrease the readability of our paper.

We agree that our article does not touch on the important subject of controversies around extracellular vesicles research. Thus, we have provided an additional section to Conclusions of our paper.

Round 2

Reviewer 1 Report

Comments and Suggestions for Authors

The authors did not meet any of my requests.

Author Response

Dear Reviewer,

We decided to include some of the ideas provided in your review. We have added a new section 5.5 Senescence, which touches the immense topic of cellular senescence.

Sincerely,

Authors

Reviewer 4 Report

Comments and Suggestions for Authors

The authors should have provided more evidence to support their concepts which seem not to be the case. I am afraid, the review still seems a bit undeveloped.

Author Response

Dear Reviewer,

After your answer, we have decided to delve into the topic of Clinical Trials and include some of them currently being conducted. We have added some new references 138-140 and prepared a table showing recent studies examining the role of extracellular vesicles in a wide perspective. We hope that after these changes our manuscript will be suitable for publication.

Sincerely,

Authors

Round 3

Reviewer 1 Report

Comments and Suggestions for Authors

accepted in the present form 

Reviewer 4 Report

Comments and Suggestions for Authors

Paper is now suitable for publication.